# Can Peat Amendment of Mars Regolith Simulant Allow Soybean Cultivation in Mars Bioregenerative Life Support Systems?

**DOI:** 10.3390/plants12010064

**Published:** 2022-12-22

**Authors:** Antonio Giandonato Caporale, Roberta Paradiso, Greta Liuzzi, Nafiou Arouna, Stefania De Pascale, Paola Adamo

**Affiliations:** Department of Agricultural Sciences, University of Naples Federico II, Via Università 100, Portici, 80055 Naples, Italy

**Keywords:** *Glycine max* (L.) Merr., controlled environment, bioregenerative life support systems (BLSSs), in situ resource utilization (ISRU), MMS-1

## Abstract

Higher plants will play a key role in human survival in Space, being able to regenerate resources and produce fresh food. However, the creation of a fertile substrate based on extra-terrestrial soils is still a challenge for space cultivation. We evaluated the adaptability of soybean (*Glycine max* (L.) Merr.) cultivar ‘Pr91M10′ to three substrates, the Mojave Mars regolith Simulant MMS-1, alone (R100), and in a mixture with blond *sphagnum* peat at two different volumes, 85:15 (R85P15) and 70:30 (R70P30), in plants directly sown on the substrates or transplanted after sowing on peat. The low pH of peat (4.34) allowed the mitigation of the alkalinity of the Mars regolith simulant (pH 8.86), lowering the initial pH to neutral (6.98, R85P15), or subacid to neutral (6.33, R70P30) values. Seed germination reached the highest percentage in the shortest time in the mixture of regolith simulant with 15% of peat. The cultivation substrate did not affect the plant growth and nutritional status. However, a significant interaction between the substrate and planting method was found in several growth parameters, with the highest positive effects observed in plants resulting from direct sowing on the regolith mixture with peat.

## 1. Introduction

Human exploration beyond Low Earth Orbit (LEO) will require technologies able to regenerate resources, while recycling the consumables and waste of space colonies. A feasible strategy to renew resources, by making a continual recycling system for waste, is the development of bioregenerative life support systems (BLSSs), combined with the in situ resource utilization (ISRU) [1]. BLSSs are artificial ecosystems in which appropriately selected organisms are assembled in consecutive steps of recycling by exploiting their metabolic routes, to reconvert the crew waste into nutritional biomass, oxygen, and potable water [2].

Higher plants represent the most promising bioregenerators, being able to renew air through photosynthesis, to purify water through transpiration, and to recover waste products through mineral nutrition, while providing fresh food and health benefits to the astronauts [3]. However, the substrate is a critical aspect for plant cultivation in Space, where fertile soils may be unsuitable and locally available materials could be the sole source of growing media. Therefore, in the view of long-term manned missions on Mars, the plant response to growth on the Martian regolith, in the different types of candidate crops (leafy vegetables, fruit vegetables, tuber plants), needs to be evaluated.

Since Mars regolith is not available on Earth, space research studies are conducted on regolith simulants, which tend to replicate the physicochemical and hydraulic properties of Mars regolith, analyzed in situ by rovers and robotic spacecrafts. Mojave Mars regolith Simulant (MMS-1) is produced and sold by The Martian Garden (Austin, Texas, USA). According to the mineralogical and chemical characterization of Caporale et al. [4], it mainly consists of plagioclases; this is in mixture with amorphous materials and zeolite, containing essential plant nutrients, such as K, Ca, Mg and Fe, but lacking organic C, N, P and S. It is alkaline (pH 8.86) and coarse textured, lacking an adequate water holding capacity and lacking the fine particles exerting colloidal properties [4]. Hence, the amendment of this Mars simulant with organic materials is needed to enhance its physicochemical and hydraulic properties, and the nutrient availability, to sustain the plant growth [4,5,6].

Recently, our team started a series of experiments to evaluate, by a multidisciplinary approach, the possible use of MMS-1 Mars regolith simulant as a cultivation substrate for different crops, in BLSSs [4,5,7,8,9,10]. The findings demonstrated that the amendment of MMS-1 with green compost [4,5,10] or horse/swine manure [7,8,9] can make simulant-amendment mixtures that are proper substrates for lettuce [4,5,7,8,9] and potato [10] cultivation; this also solves the issue of disposal of organic effluents in long term manned missions. A sustainable crop cultivation in Mars BLSSs should foresee the turnover of plant species belonging to different botanical families that can be the source of fibers (e.g., lettuce), carbohydrates (e.g., potato) and proteins (e.g., soybean) for crew diet.

Soybean [*Glycine max* (L.) Merr.] is a candidate species for hydroponic cultivation in BLSS [11,12], because of the high nutritional value of the seeds, which are an important source of protein (37–45%), lipids, dietary fiber, and biologically active substances such as isoflavones. It has been demonstrated that the presence of soy-proteins in the diet helps hamper several detrimental phenomena which are frequent in animals exposed to weightlessness, such as osteoporosis, muscle atrophy and brain damage. Soybean seeds can be eaten as sprouts after germination or transformed into several edible products, including soymilk and okara. Soymilk is a liquid extract obtained by a relatively low technology, which can complement or replace animal milk (e.g., in vegetarian diet). The co-product of soymilk extraction, named okara or soy pulp, is rich in dietary fiber (50–60%), protein and lipids, and contains significant levels of B vitamins [13]. It is used in both human and animal nutrition, to partially replace wheat flour for bread preparation, and as a fermentation stock for production of seasonings, spices, and tempeh, as well as a functional dietary additive in biscuits and snacks.

In a first attempt at greenhouse cultivation of a pot of soybean on the Mars MMS-1 regolith simulant, alone and amended with green compost at 30% in volume (characterized by Caporale et al., 2020 [4]), we observed serious symptoms of deterioration (weakness, yellowing, leaf falling) during the first developmental stages (seed germination and plantlets establishment). This was due to the alkaline pH (8.86), low content of promptly-bioavailable nutrients, but high Na bioavailability, predominance of macro vs. micropores, and consequent water deficiency [4,7]. In a recent experiment with *Arabidopsis thaliana*, grown on samples returned from the Apollo missions 11, 12, and 17, Paul et al. [14] observed a slow growth of plants on lunar regolith, with severe stress morphologies. In addition, these plants differentially expressed genes indicating ionic stresses, similar to plant reactions to salt, metal and reactive oxygen species.

Our unsatisfactory first results regarding soybean cultivated on the MMS-1-based growth media, clearly highlighted the agronomic limits of the nutrient-poor and alkaline substrates, which need further adjustments to assure soybean plant adaptation. Therefore, with the goal being to adjust the alkaline pH of MMS-1, we mixed it with blond *sphagnum* peat (an acid amendment) at two different rates (*v*:*v*), 85:15 (R85P15) and 70:30 (R70P30). Afterward, we monitored the plant adaptation of the soybean cultivar ‘Pr91M10′ in these substrates, in a pot under laboratory indoor conditions and during the first stages of development (seed germination and plantlets establishment); this was as a function of two planting procedures: (i) direct sowing on the different substrates (S), and (ii) transplanting after germination on peat (T). We used peat to exploit its acidic chemical properties, with the awareness that it cannot be a possible candidate amendment for space farming. Although peat is still widely used in horticulture, it is classified as a slowly renewable biomass, having a natural production rate of 1 mm per year. In the last years, many studies have focused on the environmental concerns over the rapid depletion of peat under global warming, with the need to find similar but sustainable alternatives [15].

Seed germination is a crucial step of plant development; it includes all the events, beginning with the water uptake by the dry quiescent seed and ending with the root apex protrusion out of the seed coat, as the result of the embryonic axis elongation. The tight control of seed germination is fundamental for both the plant establishment and the derived food characteristics; hence, the monitoring of the first stages of development (seed germination and plantlets establishment) of a selected cultivar is the first point to be addressed in candidate crops performance on plant growth substrates.

## 2. Results

### Plant Growth

The blond *sphagnum* peat used in our study (purchased by Vigorplant srl Italia, https://www.vigorplant.com (accessed on 1 June 2022)) is sold as an Irish peat derived from the slow transformation, in the absence of oxygen, of *sphagnum* and marsh mosses. It showed an acidic pH (4.34) and a low electrical conductivity (EC, 0.30 dS m^−1^), measured in 1:2.5 solid/aqueous extract. It mainly consisted of organic C (46% of the total), but it was also an important source of N (0.7%) and other key nutrients, such as K, Ca, Mg, P and Fe (0.03–0.5% each). The low pH of this peat allowed us to adjust/correct the alkaline pH of the pure MMS-1 simulant (i.e., 8.86). In fact, the addition of 15% in volume of peat to MMS-1 (R85P15) adjusted the pH of the growth medium to neutrality (pH 6.98), while the supply of peat at 30% in volume (R70P30) lowered the pH to 6.33. The EC of R85P15 and R70P30 was 0.17 and 0.21 dS m^−1^, respectively, values indicating no accumulation of soluble salts.

The percentage of germination and the Mean Germination Time (MGT) of seeds of the soybean cultivar ‘Pr91M10′ after 9 days were significantly affected by the sowing media, ranging from 45.8% in 5.6 days in pure peat, to 75.0% in 4.2 days in pure regolith simulant (R100), 100% in 3.0 days in R85P15 and 87.5% in 4.4 days in the mixture R70P30.

The planting method significantly influenced the plant growth, with higher average values of dry matter accumulation in both the aerial part and the roots recorded in plants growing from direct sowing on the tested substrates, compared to those transplanted after germination on peat (Table 1). Within each planting method, the cultivation substrate affected the plant growth differently (Table 1). Indeed, a significant interaction between the substrate and planting method was found in several growth parameters (i.e., plant height, root fresh weight, dry weight of aerial part and roots) (Table 1). In particular, the total dry biomass was higher in plants on both of the regolith mixtures with peat in the case of direct sowing (R85P15-S, R70P30-S), while it was greater on the pure regolith simulant when plantlets were transplanted (R100-T). On average, with both planting methods, the cultivation substrate seemed not to influence the plant growth significantly, because of the high variability of data.

As observed in the growth parameters, the dry matter partitioning in the plant aerial part and roots did not change in the different substrates and planting method (86.3% and 13.7%, and 88.1% and 11.9% on average in direct sowing and transplanting, respectively).

Regarding the content of carbon, nitrogen, and sulfur (CNS) in soybean biomass, we observed no statistically significant differences due to the cultivation substrate or planting method. We also noticed a better nutritional status for C and N in the aerial biomass than in the plant roots; the opposite was observed for S (Table 2). A significant interaction between the substrate and planting method was only found in shoot N content.

## 3. Discussion

Many factors, including planting methods and the cultivation substrate, considerably affect soybean seeds germination, growth and development in the early stages [16]. In particular, an adequate germination occurs when favorable conditions, such as water and oxygen supply, and appropriate temperature, are guaranteed [17]. The soybean cv ‘Pr91M10′ germination percentage, assessed on wet filter paper and incubated in the dark at the optimal environmental conditions according to the International Rules for Seed Testing (edition 1999) [18], was 93.5% and the MGT was 4.2 days [11]. In our experiment, the germination of soybean seeds was greatly affected by the substrate composition. Specifically, the lowest germination percentage and the slowest rate were observed in pure peat (45.8%). According to the disclosures of the manufacturer (Vigorplant srl Italia, https://www.vigorplant.com accessed on 20 July 2022) and our measurements, the blond *sphagnum* peat used in our study is characterized by a high content of organic matter (~80%) and lignin, an acid pH (4.34) and low EC (0.30 dS m^−1^), good water holding capacity and porosity. The reduced germination obtained in peat could be mainly due to its acid pH, since soybean is sensitive to soil acidity and negative effects are known to decrease the pH, starting from 6 [15]. On the other hand, this low pH of peat was very helpful to neutralize the alkaline pH of the pure MMS-1 simulant, when amended with 15% (R85P15, pH 6.98) or 30% (R70P30, pH 6.33) in volume of peat. The alkaline pH of pure Mars regolith simulant caused a low germination of soybean seeds (75.0%), together with all the other negative physico-hydraulic and chemical properties of this mineral substrate [4]; this is different to what is reported by Paul et al. [14], whose study obtained a ~100% of germination rate of *Arabidopsis thaliana* in all tested Apollo lunar regolith samples and JSC-1A simulant.

Clearly, germination was promoted on regolith simulant when amended with peat, since blending improved its physicochemical properties and the overall fertility. To this purpose, the addition of 15% in volume of peat seems to be the best dose for the germination process of soybean seeds.

Unexpectedly, our data showed no significant difference in the growth parameters and CNS nutritional status, recorded in plantlets 20 days after sowing on the three tested substrates, suggesting that the substrate alone has little effect on soybean seedling development during the early stages. On the other hand, Paul et al. [14] found that mature Lunar regolith, such as that from Apollo 11, provided a poorer substrate for *Arabidopsis thaliana* growth than immature regolith (i.e., from Apollo 17). As our experiment was only a preliminary test, our measurements do not enable us to explain the reason for the non-significant effect of the substrate on soybean growth; however, we can hypothesize that seedling requirements, in terms of nutrients, were still fulfilled by the reserve materials present in the seed or that all the substrates provided the small amount of water and minerals required during this stage [19].

Our data demonstrate that the planting method greatly affects the seedling establishment in soybean. Direct sowing and transplanting are known to influence the growth and development, as well as the quality of maize, wheat, sorghum, and pearl millet [20]. Although direct sowing is an economical technique, transplanting is considered more advantageous because it prevents several problems, including a low germination rate, and allows the selection of healthy seedlings. In our experiment, the higher dry matter accumulation in both the aerial part and the roots was observed in plants from direct sowing. This result is not surprising for soybean, as it is known that plantlets are not very resistant to transplanting stress [21].

## 4. Materials and Methods

### 4.1. Plant Growth Conditions and Measurements

The experiment was carried out at the Laboratory of Soil/Agricultural Chemistry of the Department of Agriculture of the University of Naples Federico II (Naples, Italy), from 15 June to 5 July 2022.

Forty-eight seeds of soybean [*Glycine max* (L.) Merr.] cultivar ‘Pr91M10′ (Pioneer^®^, https://www.pioneer.com (accessed on 1 October 2022)) were divided in two equal plots (24 seeds each). One plot was sown on pure blond *sphagnum* peat (pH 4.34, electrical conductivity 0.30 dS m^−1^) in polystyrene seed planting trays (alveolar volume: 40 mL). The other plot was sown directly in 0.35 L pots (8 × 8 × 8 cm), on Mojave Mars Simulant MMS-1 (R100-S) alone, and in a mixture with peat at two different rates (*v*:*v*), 85:15 (R85P15-S) and 70:30 (R70P30-S). At the stage of the first true leaves, plantlets obtained on pure peat were transplanted on the same substrates, to obtain another 3 treatments: R100-T, R85P15-T andR70P30-T. Based on the small plant size expected at the end of the experiment and the limited availability of the regolith simulant, 2 seeds were used per each pot (4 pots per substrate, 8 seeds per substrate in total).

Plants were grown under solar light (photoperiod 15 h) and at room temperature for 3 weeks, corresponding to the stage of second true leaves. Air temperature and relative humidity were monitored every ten minutes with a RoHS wireless Bluetooth data logger. The average day/night temperature and the relative humidity recorded in the 20 days of experiment were 29.0 ± 2.1/27.4 ± 1.3 °C and 62.1 ± 8.6/65.5 ± 8.8 %, respectively (Mean Value ± Standard Deviation).

Fertigation was performed using a nutrient solution based on the standard Hoagland [22], with the recipe half-strength modified by Wheeler et al. (2008) [23], according to the specific requirements of soybean. The pH of the nutrient solution and EC were 5.8 and 2.0 dS m^−1^, respectively. Every three pulses, fertigation was alternated with one irrigation with deionized water in order to prevent salt accumulation in the substrate. The pH and EC of the substrates were measured on dry extract at the beginning of the experiment.

The seed germination percentage and the Mean Germination Time (MGT) were evaluated at the end of the test (8 days). The MGT was calculated by counting the number of germinated seeds every day and according to the following formula: MGT= Σ (n × g)/N, where: n = number of germinated seeds per day, g = number of days from the begin of the test, N = total number of germinated seeds. The plant growth rate (plant height, number of leaves, total dry matter) during the first stages of development were recorded two times per week, in both directly sown and transplanted plants. At the end of the experiment, 20 days after sowing (DAS), the fresh weight (FW) and the dry matter accumulation (DM, after oven drying at 80 °C until constant weight) of both the plant aerial part and roots were determined with an analytic balance (Gibertini Europe 500).

For the assessment of CNS total content in soybean shoots and roots, 2 mg of dried biomass samples (n = 4) underwent combustion analysis by a Micro Elemental Analyser—UNICUBE^®^ (Elementar, Hesse, Germany), equipped with a thermal conductivity detector (TCD), to determine total C, N and S concentrations. Instrument calibration, a check of the element accuracy, and recovery were performed using a sulphanilamide (Elementar, 99.5%) standard.

### 4.2. Statistical Analysis of Data

Each combination *Substrate x Propagation method* consisted of 8 replicates (2 plants per pot in 4 pots) randomly distributed on a bench. Data were analyzed by a two-way analysis of variance (ANOVA) using the SPSS 28 software package (www.ibm.com/software/analytics/spss (accessed on 20 July 2022)), and means were compared by a Duncan post hoc test, performed at a significance level of *p* ≤ 0.05.

## 5. Conclusions

Although the substrate alone did not influence the seedling performance, interaction with the planting method greatly affected many growth parameters, with the best performance in plants resulting from direct sowing on both regolith mixtures with peat. These results suggest that direct sowing, in combination with the regolith amendment and an acid blond peat, is the best strategy in regard to the growth and development of soybean during the first stages, in light of cultivation in a controlled environment in BLSSs. However, although the use of peat allowed the study to bring the pH of growth media to optimal values, numerous negative physico-hydraulic and chemical features of the substrates still hinder the full development of soybean plants, cause strong abiotic stresses, and make the reaching of the reproductive phase impossible. Therefore, further studies and efforts to make MMS-1-based growth media suitable for soybean cultivation are still needed.

## Figures and Tables

**Table 1 plants-12-00064-t001:** Main growth parameters in plants of soybean cultivar ‘Pr91M10′ grown on Mojave Mars Simulant MMS-1 (R100) alone and in mixture with peat at two different rates (*v*:*v*), 85:15 (R85P15) and 70:30 (R70P30), from direct sowing on the different substrates and from transplanting after germination on peat. 20 DAS; Mean values, n = 4; Different letters indicate significant differences within each column (*p* < 0.05); ns, *, ** indicate non-significant or significant difference at *p* < 0.05 or 0.01, respectively (two-way ANOVA, Duncan’s multiple-range test).

	Plant Height	Fresh Weight			Dry Weight			Dry Matter
		Aerial Part	Roots	Total	Aerial Part	Roots	Total	
	(cm)	(g Plant^−1^)	(g Plant^−1^)	(g Plant^−1^)	(g Plant^−1^)	(g Plant^−1^)	(g Plant^−1^)	(% of FW)
Sowing								
R100-S	23.25 c	0.78	0.19 c	0.97	0.13 cd	0.02 bc	0.14 cd	15.08
R85P15-S	34.13 ab	1.17	0.34 abc	1.51	0.20 ab	0.04 a	0.23 a	15.45
R70P30-S	36.63 a	1.36	0.32 abc	1.68	0.21 a	0.04 a	0.25 a	14.85
mean	31.33 a	1.10	0.29	1.39	0.18 a	0.03 a	0.21 a	15.13 a
Transplanting								
R100-T	33.00 ab	1.42	0.43 a	1.85	0.19 ab	0.03 ab	0.22 ab	11.73
R85P15-T	27.75 bc	1.63	0.25 bc	1.88	0.16 bc	0.02 bc	0.17 bc	11.93
R70P30-T	20.33 c	0.89	0.36 ab	1.25	0.10 d	0.01 c	0.12 d	9.97
mean	27.03 b	1.31	0.35	1.66	0.15 b	0.02 b	0.17 b	11.21 b
Significance								
Substrate (S)	ns	ns	ns	ns	ns	ns	ns	ns
Planting method (Pm)	*	ns	ns	ns	*	**	**	**
SxPm	**	ns	**	ns	**	**	**	ns

**Table 2 plants-12-00064-t002:** Elemental analysis (C, N, S, and C/N ratio) in shoots and roots of soybean cultivar ‘Pr91M10′ grown on Mojave Mars Simulant MMS-1 (R100) alone and in mixture with peat at two different rates (*v*:*v*), 85:15 (R85P15) and 70:30 (R70P30), from direct sowing on the different substrates and from transplanting after germination on peat. 20 DAS; Mean values, n = 4; ns, ** indicate non-significant or significant difference at *p* < 0.01, respectively (two-way ANOVA, Duncan’s multiple-range test).

	C (%)		N (%)		C/N		S (%)	
	Shoots	Roots	Shoots	Roots	Shoots	Roots	Shoots	Roots
Sowing								
R100-S	42.7	33.1	7.54 a	5.10	5.66	6.50	0.55	0.81
R85P15-S	42.5	41.9	5.66 c	4.02	7.50	10.4	0.76	0.84
R70P30-S	42.4	30.6	5.42 c	2.79	7.83	11.0	0.42	1.12
mean	42.5	35.2	6.21	3.97	6.99	9.30	0.58	0.93
Transplanting								
R100-T	42.5	35.7	5.49 c	5.41	7.74	6.60	0.47	1.09
R85P15-T	45.7	37.2	6.76 b	4.54	6.75	8.18	0.42	0.65
R70P30-T	44.9	35.4	7.69 a	4.76	5.84	7.45	0.53	0.68
mean	44.4	36.1	6.65	4.91	6.78	7.41	0.47	0.81
Significance								
Substrate (S)	ns	ns	ns	ns	ns	ns	ns	ns
Planting method (Pm)	ns	ns	ns	ns	ns	ns	ns	ns
SxPm	ns	ns	**	ns	ns	ns	ns	ns

## Data Availability

The data presented in this study are available on request from the corresponding author (R.P.).

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
