# Peer review of "Can Peat Amendment of Mars Regolith Simulant Allow Soybean Cultivation in Mars Bioregenerative Life Support Systems?"

_plants, 2022, doi:10.3390/plants12010064_

Round 1
Reviewer 1 Report
The response of plants, and specifically of crop plants, to the culture in alien soils, such as those of the Moon and Mars, is one of the most important challenges affecting the coming missions of human space exploration, particularly the design of the necessary Bioregenerative Life Support Systems. The manuscript by Caporale et al. approaches this problem by means of the use of a known Mars regolith simulant complemented by a peat amendment. The experiments, which continue previous works of the same laboratory, are simple, but well designed, and the conclusions obtained are solid and add relevant information.
In my opinion, the manuscript could be accepted for publication, although I have a couple of minor criticisms that the authors should address:
- 1. The use of abbreviations is excessive and, in some cases, their meaning is not explained. This is especially significant in Table 1 (FW, DW, D.M. and CNS) and in the abstract, where probably abbreviations should not be allowed, but, in any case, the meaning of CNS is not provided.
- 2. The authors should cite and discuss the paper by Paul et al.: Plants grown in Apollo lunar regolith present stress-associated transcriptomes that inform prospects for lunar exploration. Communications Biology 2022, 5, 382, doi:10.1038/s42003-022-03334-8. It reports the first (and unique up till now) experiment of plant growth in real lunar regolith. The results of the present manuscript should be compared with the results reported in this paper and discussed accordingly.
Author Response
R1 = Reviewer 1
OR = Our response
R1 = The response of plants, and specifically of crop plants, to the culture in alien soils, such as those of the Moon and Mars, is one of the most important challenges affecting the coming missions of human space exploration, particularly the design of the necessary Bioregenerative Life Support Systems. The manuscript by Caporale et al. approaches this problem by means of the use of a known Mars regolith simulant complemented by a peat amendment. The experiments, which continue previous works of the same laboratory, are simple, but well designed, and the conclusions obtained are solid and add relevant information.
OR = We thank the Reviewer 1 for this statement on our work and on the useful suggestions, which helped us to improve our manuscript.
R1 = In my opinion, the manuscript could be accepted for publication, although I have a couple of minor criticisms that the authors should address:
- 1. The use of abbreviations is excessive and, in some cases, their meaning is not explained. This is especially significant in Table 1 (FW, DW, D.M. and CNS) and in the abstract, where probably abbreviations should not be allowed, but, in any case, the meaning of CNS is not provided.
OR = The abbreviations in Abstract and Table 1 have been removed and the meaning of CNS has been added.
- 2. The authors should cite and discuss the paper by Paul et al.: Plants grown in Apollo lunar regolith present stress-associated transcriptomes that inform prospects for lunar exploration. Communications Biology 2022, 5, 382, doi:10.1038/s42003-022-03334-8. It reports the first (and unique up till now) experiment of plant growth in real lunar regolith. The results of the present manuscript should be compared with the results reported in this paper and discussed accordingly.
OR = We thank the reviewer for this suggestion. We cited and discussed the paper in both the Introduction and Discussion sections. We also compared our germination rates with those described in the suggested paper.

Author Response
R2 = Reviewer 2
OR = Our response
R2 = The implementation of bioregenerative life-support systems is required for long-term human space exploration. The authors have conducted a series of experiments for the purpose. In this study, they examined peat amendment of Mars regolith simulant in soybean cultivation and obtained some interesting results. However, the experiment is just preliminary and the manuscript seems to be immature at the present form.
OR = We thank the reviewer for reading our work carefully and providing comments and criticisms which helped us to improve the manuscript. More than the list of the suggested corrections, we tried to make clearer the importance of our preliminary results in the frame of the overall objective of our researches.
R2 = The addition of peat to MMS-1 caused some positive effects on germination and early plant growth at direct sowing, although the cultivation substrate did not significantly influence the plant growth on the average of two planting methods. This is important results and should be confirmed by additional experiments. If the positive effects of peat addition is confirmed, whether the effect is caused by pH adjustment capacity of peat needs to be examined and confirmed. The pH adjustment by some buffering fertilizer, in place of peat, may be effective for the purpose.
OR = The brief experiment described in the paper has to be considered a preliminary test, which appeared to be a necessary step to better set up additional and more complete experiments on soybean grown on regolith-based substrates, and is only a part of a series of researches and scientific papers on the topic. In general, in our experiments, peat, green compost and manure, used as acidic matrix to improve the pH of the regolith and the pH-related properties, aim also to simulate the organic waste produced during a Mars mission (urine and faeces, crop residues) to be used as organic amendment. Some references of our works on this topic are listed:
Caporale A.G., Amato M., Duri L.G., Bochicchio R., De Pascale S., Simeone G.D.R., Palladino M., Pannico A., Rao M.A., Rouphael Y., Adamo P. (2022). Can Lunar and Martian Soils Support Food Plant Production? Effects of Horse/Swine Monogastric Manure Fertilisation on Regolith Simulants Enzymatic Activity, Nutrient Bioavailability, and Lettuce Growth. Plants, 11, 3345, https://doi.org/10.3390/plants11233345
Caporale A.G., Palladino M., De Pascale S., Duri L.G., Rouphael Y., Adamo P. (2023). How to make the Lunar and Martian soils suitable for food production - Assessing the changes after manure addition and implications for plant growth. Journal of Environmental Management 325, 116455, https://doi.org/10.1016/j.jenvman.2022.116455
R2 = The following points in the manuscript need to be revised.
- The data should be presented in the order of peat (or MMS-1) ratio, R100, R85P15, and R70P30, throughout the text, Figures and Tables.
OR = The order of the treatments has been modified in the text, Figures and Tables according to the suggestion.
- Line 17: (6.33, R85P15) should read (6.33, R70P30).
OR = We corrected the mistake in the abstract.
- Figure 1 should be deleted, because there is no significance and the description in the text is enough.
OR = We thank the reviewer for this comment. The figure has been removed.
- Ref No. 2 and 3 need page information.
OR = The references No. 2 and 3 have been completed.

Reviewer 3 Report
Manuscript "Can peat amendment of Mars regolith simulant allow soybean cultivation in Mars bioregenerative life support systems?" by Authors: Antonio Giandonato Caporale, Roberta Paradiso, Greta Liuzzi, Nafiou Arouna, Stefania De Pascale, Paola Adamo considers an important aspect of the possibility of using soil that imitates the wind-eroded surface layer of Mars regolith for growing plants. Unfortunately, the work is done and framed in such a way that it is not possible to evaluate the results of the experiments.
Firstly, there are no indications or references in the article by what criteria this soil sample is similar to regolith from the planet Mars. There are no indications for the analysis of the chemical composition, mechanical properties. There are also no references to works on the study of the properties of such samples. Also poorly characterized is the source and properties of the peat, which appears to be an example of a low-grade high-moor peat that is used to cultivate acid-demanding plants such as blueberries or heathers.
It is interesting that the authors do not consider the features and problems of peat delivery, its cost, and even more interesting that they consider peat probably only as a deoxidizer. It is surprising that other functions of this addition to the growing substrate were completely ignored and, accordingly, not taken into account in the choice of controls (for example, does peat contain some nutrients and how much?, is peat able to increase the water-holding capacity of the substrate?, chemical interaction led to the modification of regolith fragments ?...)
The system of sampling and setting controls looks no less strange, even taking into account the ignorance of the above issues. For example, if we consider the role of peat as an acidifier, then the question arises why acid solutions (for example, trivial hydrochloric acid) were not used. This question is important in two aspects at once - they should have been used to determine the amount of acid needed to change the pH, and they should have been used to test the effects of regolith with a changed pH. On the other hand, it is not clear why the seedlings could not be grown on peat, in which the pH is changed to match the pH of the regolith and the mixtures studied as substrates.
As other controls, the authors could use any other substrate that gives similar effects, such as coconut or even just cellulose (in the form of sawdust).
It is not at all clear what the authors mean by natural light (what characteristics such light has, how long the day lasts, how long the night). It can be assumed that fluorescent lighting was meant, although it is possible that the light from the window ... this does not make it possible to understand and appreciate the work.
No data on a preliminary analysis of the germination of seeds is given in the article, and the sample for such experiments seems to me insufficient, especially given the lack of repetitions, both technical and biological (at least their descriptions).
However, in addition to these shortcomings, there are also technical shortcomings that are better taken into account when revising this article. For example, in the introduction it is necessary to discuss the issues of creating these substrates, the issues of their delivery, the substrates that were used by other authors, the problems associated with the development of the root system under conditions of changed pressure, a different composition of gases.
Without photographs and analysis of the development of the aboveground and intrasubstrate parts of plants, it is difficult to judge the suitability of this substrate.
Graphs and tables do not make it possible to understand the differences with the control grown on acidic peat, and the authors did not use peat with normal (slightly acidic) pH.
In its present form, it seems to me extremely difficult to correct the work in the major revision mode, and it is with great regret that I recommend rejecting the publication of this work in the presented version.
Author Response
R3 = Reviewer 3
OR = Our response
R3 = Manuscript "Can peat amendment of Mars regolith simulant allow soybean cultivation in Mars bioregenerative life support systems?" by Authors: Antonio Giandonato Caporale, Roberta Paradiso, Greta Liuzzi, Nafiou Arouna, Stefania De Pascale, Paola Adamo considers an important aspect of the possibility of using soil that imitates the wind-eroded surface layer of Mars regolith for growing plants. Unfortunately, the work is done and framed in such a way that it is not possible to evaluate the results of the experiments.
OR = We thank the reviewer for reading our work carefully and providing comments and criticisms which helped us to improve the manuscript. More than the list of the suggested corrections, we tried to improve all the points marked in the review form as “Must be improved”, and to make clearer the importance of our preliminary results in the frame of the overall objective of our researches.
R3 = Firstly, there are no indications or references in the article by what criteria this soil sample is similar to regolith from the planet Mars. There are no indications for the analysis of the chemical composition, mechanical properties. There are also no references to works on the study of the properties of such samples. Also poorly characterized is the source and properties of the peat, which appears to be an example of a low-grade high-moor peat that is used to cultivate acid-demanding plants such as blueberries or heathers.
OR = In the Introduction, we explained that commercial Mars simulant are assembled to replicate the physicochemical and hydraulic properties of Mars regolith analyzed in situ by rovers and robotic spacecrafts, and can be effectively used in the space research studies on the Earth. We cited a couple of our papers (https://doi.org/10.1016/j.scitotenv.2020.137543 and https://doi.org/10.1016/j.jenvman.2022.116455) in which we provided the mineralogical, chemical and spectroscopic characterizations of MMS-1 simulant produced by The Martian Garden, pure or amended with organic materials similar to those that could be produced in space BLSSs. We also provided more details on peat source and chemical properties.
R3 = It is interesting that the authors do not consider the features and problems of peat delivery, its cost, and even more interesting that they consider peat probably only as a deoxidizer. It is surprising that other functions of this addition to the growing substrate were completely ignored and, accordingly, not taken into account in the choice of controls (for example, does peat contain some nutrients and how much?, is peat able to increase the water-holding capacity of the substrate?, chemical interaction led to the modification of regolith fragments ?...)
OR = In the Introduction, we stated that we used peat to exploit its acidic chemical properties, with the awareness that it cannot be a possible candidate amendment for space farming. Although peat is still widely used in horticulture, it is classified as slowly renewable biomass, having a natural production rate of 1 mm per year. In the last years, many studies focused on the environmental concerns over the rapid depletion of peat under global warming, with the need to find similar but sustainable alternatives. We agree with Reviewer 3 regarding the importance of substrate characterization to explain and discuss crop growth performance. In fact, we basically adopted this approach in our previous published experiments on the topic. In this case, we are just describing preliminary results in a Brief Report. Certainly, we will deepen our studies with a complete characterization of MMS-1/peat mixtures in the near future.
R3 = The system of sampling and setting controls looks no less strange, even taking into account the ignorance of the above issues. For example, if we consider the role of peat as an acidifier, then the question arises why acid solutions (for example, trivial hydrochloric acid) were not used. This question is important in two aspects at once - they should have been used to determine the amount of acid needed to change the pH, and they should have been used to test the effects of regolith with a changed pH. On the other hand, it is not clear why the seedlings could not be grown on peat, in which the pH is changed to match the pH of the regolith and the mixtures studied as substrates.
OR = Our experiment was not set up in a hydroponic system, hence, we believe that the use of acid solutions (in particular those containing hydrochloric acid) to adjust MMS-1 alkaline pH, could have caused severe physiological problems to the plants and rhizosphere biota. We did not use pure peat as growth medium (but just as a germination substrate for seedlings to transplant in MMS-1/peat mixtures), because space farming cannot be conceived and developed on the use of a non-sustainable 100%-organic material substrate. Along with water and oxygen, recycled and stabilized organic matter is another limited resource in a space BLSSs.
R3 = As other controls, the authors could use any other substrate that gives similar effects, such as coconut or even just cellulose (in the form of sawdust).
OR = We will take into consideration the alternative substrates proposed by the reviewer 3 for future experiments. Anyway, as stated earlier, we can use them just as amendment of pure MMS-1 (to mix up to 20-30% in volume), and not as non-sustainable pure control.
R3 = It is not at all clear what the authors mean by natural light (what characteristics such light has, how long the day lasts, how long the night). It can be assumed that fluorescent lighting was meant, although it is possible that the light from the window ... this does not make it possible to understand and appreciate the work.
OR = Plantlets were exposed to solar light, under the natural photoperiod of the month of July. This has been clarified in the current version of the manuscript.
R3 = No data on a preliminary analysis of the germination of seeds is given in the article, and the sample for such experiments seems to me insufficient, especially given the lack of repetitions, both technical and biological (at least their descriptions).
OR = We thank the reviewer for highlighting this weakness about germination data. We have now added references for germination performance of the same soybean cultivar assessed by our team in a previous experiment, according to the international rules of seed testing (i.e., Paradiso R., Buonomo R., De Micco V., Aronne G., Palermo M., Barbieri G., De Pascale S., 2012. Soybean cultivar selection for Bioregenerative Life Support Systems (BLSSs). Hydroponic cultivation. Advances in Space Research, 50, pp. 1501-1511. http://dx.doi.org/10.1016/j.asr.2012.07.025). We agree that, in order to make scientific results reliable, a proper number of samples and replicates need to be used. We made this mistake in the view of performing a preliminary test, to be perfected in subsequent experiments.
R3 = However, in addition to these shortcomings, there are also technical shortcomings that are better taken into account when revising this article. For example, in the introduction it is necessary to discuss the issues of creating these substrates, the issues of their delivery, the substrates that were used by other authors, the problems associated with the development of the root system under conditions of changed pressure, a different composition of gases.
OR = We thank the reviewer for highlighting this weakness in the Introduction. Unfortunately, it seemed to us that discussing all the issues related to cultivation substrates in Space was not feasible for a single paper, and even more for a brief report. For this reason, in Introduction we only summarized the context of plant cultivation in BLSSs and the importance of IRSU, then we focused our foreword on the most relevant information about regolith simulants and soybean as selected candidate crops, even though, being our team involved in research on several of the issues listed by the referee, we are aware of their crucial importance.
R3 = Without photographs and analysis of the development of the aboveground and intrasubstrate parts of plants, it is difficult to judge the suitability of this substrate.
Graphs and tables do not make it possible to understand the differences with the control grown on acidic peat, and the authors did not use peat with normal (slightly acidic) pH.
In its present form, it seems to me extremely difficult to correct the work in the major revision mode, and it is with great regret that I recommend rejecting the publication of this work in the presented version.
OR = As stated earlier, in this Brief Report we are basically describing preliminary results of a basis horticultural study. Based on these few findings, we intend to set up and carry out a more complete experiment, in the near future, to better understand the properties and fertility of sustainable candidate substrates for space farming and the related crop productivity and health.

Round 2
Reviewer 2 Report
The revised version is clearly improved and I don't oppose its acceptance as the preliminary report. The confirmation of results by the author's future study has great expectations.
Reviewer 3 Report
The manuscript "Can peat amendment of Mars regolith simulant allow soybean cultivation in Mars bioregenerative life support systems? by Antonio Giandonato Caporale, Roberta Paradiso , Greta Liuzzi, Nafiou Arouna, Stefania De Pascale, Paola Adamo in this edition can be printed as it is promising brief report, it can be assumed that in the future the authors will expand their study and this will make it possible to clarify the currently hidden obstacles to plant cultivation in the regolith of Mars.
I think that the complexity of the task and the non-standard approach will help to reveal the possibilities of this original research.